# The Neutrophil to Lymphocyte Ratio and Other Full Blood Count Indices in Retinal Diseases: A Systematic Review of the Literature

**DOI:** 10.3390/medicina61010125

**Published:** 2025-01-14

**Authors:** Dimitrios Kazantzis, Genovefa Machairoudia, Panagiotis Theodossiadis, Irini Chatziralli

**Affiliations:** 2nd Department of Ophthalmology, Attikon Hospital, National and Kapodistrian University of Athens, 12462 Athens, Greece; genevievemach@icloud.com (G.M.); patheo@med.uoa.gr (P.T.); eirchat@med.uoa.gr (I.C.)

**Keywords:** neutrophil to lymphocyte ratio, complete blood count, age-related macular degeneration, diabetic retinopathy, retinal vein occlusion

## Abstract

*Background and Objectives:* The neutrophil to lymphocyte ratio (NLR) and other full blood count indices have been used as a marker of inflammation in a variety of diseases. The aim of the current review is to summarize the existing knowledge on the use of these indices in retinal diseases. *Materials and Methods:* A systematic review of the literature was conducted to find eligible articles. The PUBMED and Scopus databases were systematically searched for relevant studies examining full blood count indices in retinal diseases. *Results*: The NLR was elevated in a number of vitreoretinal conditions, such as wet age-related macular degeneration (AMD), diabetic retinopathy and retinal vein occlusion, compared to controls. Full blood count indices could be useful in predicting the response to anti-VEG treatment in patients with wet AMD or diabetic macular edema (DME). *Conclusions*: The NLR and other indices can be used as diagnostic markers in retinal diseases and as prognostic factors of the response to treatment. The small sample size and short follow-up of the included studies and the variation in the measurement and cutoffs used for the NLR are limitations of its use in retinal conditions. Future studies need to further validate these findings and try to establish a link between these ratios and retinal phenotypes.

## 1. Introduction

Retinal disorders consist of a variety of conditions that affect the retina and could lead to a significant and irreversible loss of vision. Although this group of conditions is heterogenous, inflammation has been associated with a number of retinal diseases, such as age-related macular degeneration, diabetic retinopathy and retinal vein occlusion [1,2]. Systemic inflammation can be evaluated in routine blood tests that provide information about cell counts and their ratios. The neutrophil to lymphocyte ratio (NLR), platelet to lymphocyte ratio (PLR) and other full blood count-derived ratios are inexpensive and readily available means of evaluating inflammation and have been evaluated in cardiovascular, autoimmune and inflammatory diseases, as well as in cancer [3,4,5,6]. Recently, these ratios have been applied to retinal diseases in an effort to investigate whether they could be used as biomarkers in such conditions. They reflect subclinical inflammation, which is crucial in retinal pathologies. The advantages of these ratios are that they are inexpensive, readily available and reliable, while the disadvantages include the fact that they can be increased in a wide range of conditions and are influenced by factors such as emotional stress and trauma.

While full blood count indices are used in retinal conditions, there is no consensus on their use and cutoff values. Based on the above, the aim of this review is to summarize the key findings of the association of the NLR and other full blood count indices with retinal diseases.

## 2. Methods

The PUBMED and Scopus databases were systematically searched on 30 August 2024 by two authors (DK and GM) using the keywords “retina”, “complete blood count”, “full blood count” and “neutrophil to lymphocyte ratio”. Studies were excluded if they were (1) case reports, (2) conference papers or (3) animal studies. Articles and book chapters cited in the reference lists of articles obtained through this method were reviewed and included when considered appropriate. No language restrictions were set. Relevant information pertaining to the topic of this review was included.

## 3. Results

### 3.1. Diabetic Retinopathy

Diabetic retinopathy (DR) is a common microvascular complication of diabetes mellitus (DM) affecting the retinal vessels. Inflammation is believed to play a crucial role in the development and promotion of DR severity [7]. Multiple studies have shown a strong and consistent relationship between an elevated NLR and the presence of DR in diabetic patients [8,9,10,11]. A meta-analysis pooled the available data and found that the NLR was increased in patients with retinopathy compared to diabetic patients without retinopathy [12]. Furthermore, other full blood count-derived ratios and indices, such as the platelet to lymphocyte ratio (PLR), the mean platelet volume (MPV) and the platelet distribution width (PDW), were also elevated in patients with DR. Apart from the presence of retinopathy, increased values of the NLR have been associated with the severity of DR. Ulu et al. found that the NLR correlated with the severity of DR grades (r = 0.630, *p* < 0.001), while Huang et al. found that NLR values increased in parallel with the DR severity. In addition, Atli et al. concluded that NLR and PLR values were significantly higher in patients with proliferative DR compared to non-proliferative DR [13,14,15].

Diabetic macular edema (DME) is the leading cause of vision impairment in patients with DM and inflammation has been implicated in the pathogenesis of DME [16]. The NLR has been found not only to be associated with the presence of DME but also to predict the response to anti-VEGF treatment [17,18]. Hu et al. studied 91 treatment-naïve patients with DME, treated with intravitreal ranibizumab, and concluded that the pre-treatment NLR values correlated with the best corrected visual acuity (BCVA) at the end of the follow-up, indicating that a higher pre-treatment NLR may contribute to an inferior BCVA outcome [17]. Conversely, Vural et al. found that higher pre-treatment values of the NLR were associated with better BCVA in patients with DME treated with an intravitreal dexamethasone implant [18]. Interestingly, some studies tried to evaluate whether certain imaging characteristics correlated with NLR values. Zhou et al. found that the NLR, PLR and systemic immune–inflammation index (SII) correlated with the number of hyperreflective foci (HRF), while Dimitriou et al. found that patients with DME and HRF had increased white blood counts [19,20].

Based on the above, we can conclude that patients with DR have increased values of the NLR and that the NLR is associated with complications related to DR. Therefore, the NLR could be used to predict the treatment response or guide management decisions in patients with DME.

### 3.2. Age-Related Macular Degeneration

Age-related macular degeneration (AMD) is a cause of severe vision loss and blindness, particularly in developed countries [21]. Its pathogenesis is multifactorial, and genetic, metabolic and environmental factors have been linked to the development of AMD [22]. Recent studies have found that inflammation can play a key role in the development of both dry and wet age-related macular degeneration [23]. Dry age-related macular degeneration is characterized by the presence of drusen, while wet age-related macular degeneration is characterized by choroidal neovascularization (CNV). Local inflammation leads to drusenogenesis, retinal pigment epithelial (RPE) degeneration, the disruption of the Bruch’s membrane and the development of CNV [24].

Table 1 summarizes the key findings of studies examining the NLR and other complete blood count-derived indices in patients with AMD [25,26,27,28,29,30,31,32,33,34,35,36].

The role of the NLR as a biomarker in AMD is dubious, as previous studies yielded contradictory results. Some found that the NLR values were higher in wet AMD patients compared to controls [25,28,31], while others did not find a significant change [27,29,32,34,36]. Interestingly, Hu et al. found that wet AMD patients had increased complete blood count indices compared to PCV patients [33]. Another noteworthy finding is that intravitreal aflibercept led to a reduction in the NLR post-treatment, while ranibizumab treatment also reduced the values, but this did not reach statistical significance [30]. Furthermore, Gökce et found that NLR values predicted the response to anti-VEGF treatment in patients with wet AMD, as the ROC analysis revealed that a cutoff value of 2.0 for the NLR could predict a change of at least 100 μm in the CMT with a sensitivity of 87.1% and a specificity of 87.8% and that a cutoff value of 2.4 for the NLR could predict a visual improvement of at least 0.1 logMAR with a sensitivity of 77.2% and a specificity of 64.8%, suggesting that the NLR could have a predictive role in treatment-naïve wet AMD patients [35].

### 3.3. Retinal Vein Occlusion

Retinal vein occlusion (RVO) is the second most common vascular disease after DR and a major cause of vision loss due to complications, such as macular edema and neovascularization [37]. It is divided into central retinal vein occlusion (CRVO) and branch retinal vein occlusion (BRVO) as per the site of the occlusion. The pathogenesis of RVO is multifactorial, and inflammation plays a crucial role in the development and clinical complications of the condition. RVO develops as the result of a combination of systemic changes, known as the Virchow’s triad, which include venous stasis, degenerative changes in the vessel wall and blood hypercoagulability [38]. Damage to the retinal vessel wall from atherosclerosis alters the rheological properties in the adjacent vein, contributing to the venous stasis and subsequent thrombosis. Several local and systemic inflammatory mediators, such as cytokines, are released secondary to the endothelial damage and have been found to be elevated in patients with RVO both in the aqueous humor and the vitreous [39,40]. The NLR has been used as a marker for both atherosclerosis and endothelial damage and an increasing number of studies have examined its use in RVO.

Table 2 summarizes the characteristics and key findings of studies exploring the use of full blood count-derived indices in RVO [41,42,43,44,45,46,47,48,49,50,51,52,53,54,55]. Not only were the NLR and other complete blood count indices increased in patients with RVO compared to controls, but they could be used as prognostic factors for anti-VEGF treatment in macular edema secondary to RVO and could be correlated with specific imaging characteristics and complications. Zhang et al. found that NLR values can predict the development of neovascular glaucoma (NVG) in patients with RVO [44]. Furthermore, NLR values were correlated with the mean visual field defects in patients that developed NVG. Chatziralli et al. studied 53 treatment-naïve patients with macular edema secondary to RVO and found that the monocyte to lymphocyte ratio (MLR) could predict a “favorable” response to anti-VEGF treatment [49]. Rao et al. also found that the pre-treatment PLR may be used as a predictive and prognostic tool for effective intravitreal injection treatment outcomes [54]. Kazantzis et al. found that higher NLR values were linked to subretinal fluid in spectral domain OCT (SD-OCT), while Timur et al. found that patients with RVO and subretinal detachment (SRD) in OCT imaging had an increased NLR and SII compared to patients without SRD [47,52]. Finally, Wang et al. concluded that NLR values were correlated with higher IL-6 levels and SII values were correlated with IL-6 and VEGF levels in the aqueous humor in patients with RVO [53].

### 3.4. Retinal Artery Occlusion

Retinal artery occlusion (RAO) is a serious ocular emergency that typically affects older individuals [56]. A RAO typically develops due to an embolus arising from atherosclerotic plaques in patients with carotid artery disease. Atherosclerosis is a chronic inflammatory disease affecting the vessel walls [57]. A number of studies have shown that the NLR is increased in patients with RAO compared to controls [58,59,60]. Interestingly, Elbeyli et al. found that the RDW might be a better predictor of RAO compared to the NLR and PLR [61], while Qin et al. concluded that NLR and PLR values are increased in patients with RAO compared to RVO [50].

### 3.5. Uveitis

Uveitis is a common intraocular inflammatory disease that can lead to permanent visual loss [62]. It is critical to fully investigate the etiology of the inflammation by acquiring the medical history, performing a physical examination and evaluating imaging and laboratory findings in order to classify and treat the intraocular inflammation promptly. Recent studies have speculated that intraocular inflammation might be reflected in complete blood count indices. Ozgonul et al. found an elevated NLR and PLR in patients with idiopathic acute anterior uveitis, a finding that was confirmed in patients who presented with anterior uveitis secondary to Bechet’s disease [63,64]. It is noteworthy that Lee et al. followed 114 patients with Behcet’s uveitis and found that high NLR and PLR values were correlated with a poor visual outcome [65]. Another interesting finding was that complete blood count indices could be used to differentiate between infectious and non-infectious uveitis as they tended to be higher in infectious uveitis [66].

### 3.6. Ischemic Optic Neuropathy

Ischemic optic neuropathies describe a group of diseases that affect the optic nerve and result in sudden vision loss. They include non-arteritic and arteritic ischemic optic neuropathy. Non-arteritic anterior ischemic optic neuropathy (NAION) is the most frequent form of acute optic neuropathy in individuals aged above 50 years, and local and systemic inflammation affect the pathogenesis of the condition by inducing hypercoagulability [67]. Arteritic ischemic optic neuropathy (AION) is a vision- and life-threatening condition that is caused by giant cell arteritis, a medium- and large-vessel vasculitis [68]. The erythrocyte sedimentation rate (ESR) and C reactive protein (CRP) are the subjects of two essential investigations in the diagnosis of AION, but a complete blood count is also obtained for patients with a suspicion of AION. Complete blood count indices have been found to be elevated in patients with both NAION and AION [69,70]. Inanc et al. investigated whether these indices could be used to differentiate between AION and NAION and found that NLR values were significantly higher in the AION group compared to patients diagnosed with NAION, an important finding that could be useful in a clinical setting [71].

### 3.7. Central Serous Retinopathy

Central serous retinopathy (CSR) is a retinal condition characterized by choroidal hyperpermeability, retinal pigment epithelium disruption and the accumulation of subretinal fluid [72]. Previous imaging studies have shown that the choroidal hyperpermeability might be secondary to stasis, ischemia or inflammation [73]. A few studies have evaluated inflammatory parameters in patients with CSR. In one study, the NLR values were not elevated in patients with CSR compared to controls, while another study found that the mean platelet volume was elevated in patients with CSR relative to controls [74,75]. Inflammatory molecules, such as chemokines, cytokines and adhesion molecules, are involved in the pathogenesis of CSR, but future research needs to shed light on their exact role in CSR and whether this process could be reflected in complete blood count tests [76].

### 3.8. Vitreomacular Disorders

Vitreomacular disorders, such as epiretinal membranes and macular holes, are common conditions that can lead to vision loss. An epiretinal membrane is a fibrocellular proliferation that develops on the surface of the internal limiting membrane. Histological evidence suggests that it includes glial cells, collagen, hyalocytes and fibroblasts and it develops idiopathically or secondary to inflammation or surgery [77]. NLR values were higher in patients with an idiopathic ERM compared to controls, suggesting that subclinical inflammation may accompany an ERM [78]. This finding was confirmed in patients with vitreomacular traction syndrome compared to controls, suggesting that low-grade inflammation as shown in complete count indices is involved in patients with VMTS [79]. Another important discovery was made by a recent study that evaluated 150 patients with a rhegmatogenous retinal detachment treated with a vitrectomy and 51 age- and sex-matched controls and concluded that NLR values could predict the development of proliferative vitreoretinopathy, a detrimental complication in which proliferative, contractile cellular membranes form in the vitreous and retina, resulting in tractional retinal detachment [80].

## 4. Conclusions

In this review, we tried to summarize the available information regarding the association between complete blood count indices and retinal pathologies. Neutrophils secrete various inflammatory mediators that participate in the pathogenesis of retinal disorders, such as DR, vascular occlusions and macular degeneration. Although there is no consensus on the ideal biomarkers derived from full blood counts, most studies employ the NLR and use different cutoffs to diagnose a wide range of retinal diseases that include diabetic retinopathy, age-related macular degeneration, retinal vein occlusion and central serous retinopathy. Practical considerations such as the laboratory assessment or the confounding effect of comorbidities that might exert influence on the full blood count indices must also be taken into consideration. Moreover, the NLR has been identified as a biomarker that predicts the response to anti-VEGF treatment in retinal conditions that require such treatment, a finding which could be particularly useful in a clinical setting. Limitations include the small sample size and short follow-up in the included studies and the variation in the measurement of the full blood count indices. Furthermore, we did not include a methodological quality assessment of the included studies. Future research should focus on the use of the NLR and other indices as a prognostic indicator in retinal disorders that require treatment and explore its use in predicting the different phenotypes of retinal conditions. Ideally, longitudinal studies on the integration of the NLR with imaging biomarkers and the response to treatment over time would clarify the associations between full blood count indices and imaging phenotypes and assist clinicians in their decision making.

## Figures and Tables

**Table 1 medicina-61-00125-t001:** Summary of key findings of studies examining relationship between NLR and AMD.

Author	Region	Year	Participants	Indices Analyzed	Key Findings
Ilhan	Turkey	2014	81 patients with dry AMD, 84 patients with wet AMD and 80 age- and sex-matched controls	NLR	Patients with dry and wet AMD had increased NLR compared to controls.
Kurtul	Turkey	2016	40 patients with dry AMD, 40 patients with wet AMD and 40 controls	NLR	Patients with wet AMD had higher NLR values compared to patients with dry AMD but no difference between AMD groups and controls.
Subhi	Denmark	2017	164 patients with wet AMD, 33 with early AMD, 56 with geographic atrophy and 77 controls	NLR, MXD *	No significant change in NLR values among groups; MXD counts were increased in patients with new onset of wet AMD.
Sengul	Turkey	2017	100 patients with wet AMD and 100 age- and sex-matched controls	NLR, PLR	NLR and PLR values were higher among wet AMD patients compared to controls. NLR and PLR values were found to be inversely proportional to BCVA and directly proportional to CMT.
Pinna	Italy	2018	78 patients with AMD (both dry and wet) and 78 controls	NLR, PLR, MLR, MPV	No significant changes in NLR, PLR, MLR and MPV values between patients with AMD and controls.
Erdem	Turkey	2021	24 patients with wet AMD who received ranibizumab and 25 patients with wet AMD who received intravitreal aflibercept	NLR, PLR, MLR	The NLR, PLR and MLR decreased post-treatment in the aflibercept group but not in the ranibizumab group.
Karahan	Turkey	2021	60 patients with wet AMD, 60 patients with dry AMD and 71 controls	NLR, PLR, MLR	NLR and MLR values were higher among wet AMD patients compared to controls. ROC curve analyses revealed that the area under the curve (AUC) for the NLR and MLR was 0.920 and 0.717, respectively, for wet AMD. The sensitivity and specificity of the NLR were 64% and 93%, respectively, whereas for the MLR they were 63% and 75%, respectively.
Naif	Saudi Arabia	2022	90 patients with dry AMD and 270 controls	NLR	No significant changes in NLR values between patients with dry AMD and controls.
Hu	China	2022	65 patients with wet AMD, 37 patients with PCV and 110 controls	NLR, PLR, MLR	NLR, PLR and MLR values were significantly higher in patients with nAMD compared to PCV.
Tricorache	Romania	2023	91 patients with wet AMD and 90 controls	NLR, PLR	No significant changes in NLR values between patients with dry AMD and controls.
Gökce	Turkey	2024	112 patients with wet AMD who received 3 monthly loading bevacizumab injections	NLR	ROC analysis revealed a cutoff value of 2.0 for NLR to predict at least 100 μm CMT change (sensitivity of 87.1%; specificity of 87.8%) and cutoff value of 2.4 for NLR to predict at least 0.1 logMAR visual improvement (sensitivity of 77.2%; specificity of 64.8%) after 3 monthly IVT bevacizumab injections.
Gunay	Turkey	2024	33 patients with wet AMD and 43 age- and sex-matched controls	NLR, PLR, SII	No significant changes in NLR values between patients with wet AMD and controls.

* MXD: the combined count of mixed white blood cells, which includes monocytes, eosinophils and basophils.

**Table 2 medicina-61-00125-t002:** Summary of key findings in studies that utilized NLR in patients with RVO.

Author	Region	Year	Participants	Indices Analyzed	Key Findings
Dursun	Turkey	2015	40 patients with RVO and 40 age- and sex-matched controls	NLR	Patients with RVO had increased NLR compared to controls. ROC curve analysis found an optimal cutoff of 1.87 for the NLR for the prediction of RVO with a sensitivity of 72.5% and specificity of 100%.
Kumral	Turkey	2016	30 patients with BRVO and 27 age- and sex-matched controls	NLR, Mean Platelet Volume (MPV)	NLR and MPV values did not differ significantly between BRVO patients and controls.
Şahin	Turkey	2019	111 patients with RVO and 88 age- and sex-matched controls	NLR, PLR	Patients with RVO had increased NLR and PLR values compared to controls. ROC curve analysis found that the optimal cutoff values of the NLR and PLR to predict retinal vein occlusion were 1.63 and 98.50, respectively.
Zhang	China	2019	Patients with RVO who developed neovascular glaucoma (NVG)	NLR, PLR, MLR	Logistic regression analysis found that NLR can predict NVG development in patients with RVO. Multiple linear regression analysis found that NLR correlated with mean visual field defects in patients with NVG secondary to RVO.
Zhu	China	2019	81 patients with BRVO and 81 age- and sex-matched controls	NLR, PLR	In ROC analysis, a cutoff of 2.48 for the NLR and 110.2 for the PLR predicted BRVO.
Pinna	Italy	2021	127 patients diagnosed with RVO and 127 age- and sex-matched controls	NLR, Mean Platelet Volume (MPV)	NLR values did not differ between patients and controls. Patients with RVO had higher MPV values than controls.
Kazantzis	Greece	2022	54 patients with RVO and 54 age- and sex-matched controls	NLR, PLR, SII	Patients with RVO had increased NLR, PLR and SII values compared to controls.
Zuo	China	2022	56 patients with RVO and 56 age- and sex-matched controls	NLR, PLR, SII	Patients with RVO had increased NLR and SII values compared to controls.
Chatziralli	Greece	2022	53 patients with treatment-naïve macular edema secondary to RVO	Monocyte to Lymphocyte Ratio (MLR)	MLR could predict “favorable” response to anti-VEGF treatment in patients with macular edema secondary to RVO.
Qin	China	2022	50 patients with RVO and 50 controls	NLR, PLR, MHR	Patients with RVO had increased NLR, PLR and MHR values compared to controls.
Timur	Turkey	2023	60 patients with RVO and serous retinal detachment (SRD), 60 patients with RVO without SRD and 60 controls	NLR, PLR, SII	Patients with RVO and SRD had increased NLR and SII compared to patients without SRD.
Kazantzis	Greece	2023	65 patients with treatment-naïve RVO	NLR	Eyes with subretinal fluid (SRF) exhibited significantly higher values of NLR.
Wang	China	2023	64 patients with RVO and 64 age- and sex-matched controls	NLR, SII, SIRI **	NLR, SII and SIRI values were increased in patients with RVO compared to controls and in patients with ischemic RVO compared to non-ischemic RVO. NLR values were correlated with higher IL-6 levels and SII values were correlated with IL-6 and VEGF levels in aqueous humor.
Rao	China	2023	315 patients with macular edema secondary to RVO treated with anti-VEGF injections	NLR, PLR, MLR	Higher pre-treatment PLR was associated with BCVA in patients with macular edema secondary to RVO.
Doǧan	Turkey	2023	30 patients with RVO and a serous macular detachment (SMD) and 30 patients with RVO and no SMD	NLR, PLR, SII	NLR and SII were significantly higher in patients with SMD compated to patients without SMD. The PLR values did not differ among the two groups.

** SIRI: systemic immune response index.

## Data Availability

No new data were created or analyzed in this study.

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
