# Peer review of "The Neutrophil to Lymphocyte Ratio and Other Full Blood Count Indices in Retinal Diseases: A Systematic Review of the Literature"

_medicina, 2025, doi:10.3390/medicina61010125_

Round 1
Reviewer 1 Report
Comments and Suggestions for Authors
This article explores the role of neutrophil-to-lymphocyte ratio and other complete blood count-derived indices as biomarkers.. The review summarizes studies examining NLR’s diagnostic and prognostic significance in conditions such as diabetic retinopathy, age-related macular degeneration and retinal vein occlusion .The manuscript is comprehensive and addresses a relevant topic by focusing on the utility of inexpensive and widely available biomarkers for retinal diseases. However, there are areas for improvement in its structure and presentation.
abstract: include the total number of studies reviewed and highlight key findings,
introduction: explain why NLR is particularly relevant for retinal diseases compared to other biomarkers, and briefly discuss its advantages in ocular diseases. identify a gap in the literature and disscuss it. expand the introduction regarding retinal diseases (togheter or separately), epidemiology, public health, consequences
methods: this part has major flaws. the authors need to completely revise this. they need to have: inclusion and exclusion criteria (preferably also a table with this information), the description of the data search (who did it, how was the data extracted), how did they manage the articles, what did they extract from the articles.
results: this section is well-organized but overly descriptive in some areas, making it difficult to identify key findings.
it does not address the limitations of the studies reviewed, such as small sample sizes or variability in NLR measurement methods. include a limitation part.
the conclusion is concise but somewhat generic. It does not emphasize the review’s most critical findings,
Author Response
abstract: include the total number of studies reviewed and highlight key findings,
introduction: explain why NLR is particularly relevant for retinal diseases compared to other biomarkers, and briefly discuss its advantages in ocular diseases. identify a gap in the literature and disscuss it. expand the introduction regarding retinal diseases (togheter or separately), epidemiology, public health, consequences
Response: These have been added.
methods: this part has major flaws. the authors need to completely revise this. they need to have: inclusion and exclusion criteria (preferably also a table with this information), the description of the data search (who did it, how was the data extracted), how did they manage the articles, what did they extract from the articles.
Response: More details have been added to this section to include: the authors that did the search, the database searched, the terms used, the date of the search, the inclusion and exclusion criteria and the language restrictions that were applied.
results: this section is well-organized but overly descriptive in some areas, making it difficult to identify key findings.
it does not address the limitations of the studies reviewed, such as small sample sizes or variability in NLR measurement methods. include a limitation part.
the conclusion is concise but somewhat generic. It does not emphasize the review’s most critical findings,
Response: This has been revised to be more concise and a limitation part has been added to reflect the limitations of the included studies as suggested.
Reviewer 2 Report
Comments and Suggestions for Authors
This systematic review evaluates the association between neutrophil-to-lymphocyte ratio (NLR) and other full blood count-derived indices with various retinal diseases, including diabetic retinopathy (DR), age-related macular degeneration (AMD), retinal vein occlusion (RVO), and others. The authors analyzed studies from PUBMED and Scopus databases to summarize evidence supporting the utility of these indices as markers of inflammation, predictors of disease severity, and treatment response. The findings suggest that elevated NLR and related indices are linked to retinal disorders and might serve as prognostic tools. However, variability in study designs and outcomes highlights the need for standardized research methodologies.
Major point:
- The introduction provides a clear overview of the relevance of NLR and other indices to systemic inflammation and retinal diseases. However, the authors should elaborate on the rationale for focusing on these indices specifically for retinal conditions. Why are other inflammatory biomarkers not considered as part of this review?
- The methods section outlines a systematic approach, including database searches and inclusion criteria. However, details about the quality assessment of included studies are missing. Were tools like PRISMA or ROBIS used to ensure the validity and reliability of the included articles?
- The search strategy lacks specifics about the inclusion of gray literature or unpublished studies. Addressing publication bias would strengthen the review.
- While the results are comprehensive, the authors should clarify the heterogeneity across studies in terms of population demographics, disease stages, and treatment modalities. Was a meta-analysis considered to quantify the pooled effect sizes?
- In Table 1 (AMD) and Table 2 (RVO), the studies’ methodological quality is not addressed. Including a column for bias risk or study limitations would improve interpretability.
- Contradictory findings, especially in AMD, should be discussed with potential explanations, such as differences in disease pathophysiology or treatment regimens.
- The discussion effectively synthesizes findings but could benefit from more critical analysis. For example, why do some studies show significant associations between NLR and disease severity while others do not? Are there thresholds for clinical utility?
- The authors suggest using NLR as a prognostic marker but do not address practical challenges, such as variability in laboratory standards or the influence of comorbidities. Adding this context would improve applicability.
- Future research directions are mentioned briefly. Expanding on specific gaps, such as the need for longitudinal studies or the integration of NLR with imaging biomarkers, would be helpful.
Minor opinions:
- Page 1, Abstract: “Retina” should be “retinal” in the keywords section for consistency.
- Page 4: “Drusogenesis” should be corrected to “drusenogenesis”.
- Page 9: Sentence starting with "Interestingly, some studies tried to evaluate..." has redundant phrasing and could be restructured for clarity.
- Tables: Ensure uniform formatting of table headings and consistent font size. Table 1 and Table 2 headings differ in style.
- References: Ensure consistent formatting of journal titles, with some entries missing capitalization (e.g., "Ther Clin Risk Manag" vs. "Int Ophthalmol").
- Figures and Tables: Include legends that are more detailed to facilitate standalone interpretation.
- Page 3: Missing commas in multi-clause sentences, such as “Further studies are needed to elucidate the exact role of NLR in retinal diseases and its potential integration into clinical practice.”
Comments on the Quality of English LanguageThe manuscript's English is clear enough to convey the research, but there are minor grammatical issues and repetitive phrasing that could be refined to improve clarity and readability. Enhanced editing would ensure the research is expressed more effectively.
Author Response
- The introduction provides a clear overview of the relevance of NLR and other indices to systemic inflammation and retinal diseases. However, the authors should elaborate on the rationale for focusing on these indices specifically for retinal conditions. Why are other inflammatory biomarkers not considered as part of this review?
Response: We only focused on full blood count derived indices and ratios and not other marker of inflammation such as hs-CRP as these can be easily available for any clinic or laboratory globally.
- The methods section outlines a systematic approach, including database searches and inclusion criteria. However, details about the quality assessment of included studies are missing. Were tools like PRISMA or ROBIS used to ensure the validity and reliability of the included articles?
Response: Unfortunately we did not include using such tools to asses the validity of the included studies and we cannot do it as part of the present study. We initially considered conducting a meta-analysis of the diagnostic accuracy of full blood count indices in retinal conditions but we dismissed this idea due to the heterogeneity of the conditions studies and the indices used.
- The search strategy lacks specifics about the inclusion of gray literature or unpublished studies. Addressing publication bias would strengthen the review.
Response: We have now included that relevant articles in the references were also extracted and that we didn't search grey literature such as conference papers in our methods section.
- While the results are comprehensive, the authors should clarify the heterogeneity across studies in terms of population demographics, disease stages, and treatment modalities. Was a meta-analysis considered to quantify the pooled effect sizes?
Response: We initially considered conducting a meta-analysis of the diagnostic accuracy of full blood count indices in retinal conditions but we dismissed this idea due to the heterogeneity of the conditions studies and the indices used.
- In Table 1 (AMD) and Table 2 (RVO), the studies’ methodological quality is not addressed. Including a column for bias risk or study limitations would improve interpretability.
Response: Unfortunately we cannot add this as part of the present study but we will acknowledge it as part of the limitations of the study.
- Contradictory findings, especially in AMD, should be discussed with potential explanations, such as differences in disease pathophysiology or treatment regimens.
Response: We tried to explain the contradictory finding of NLR levels being increased in some studies and normal in others but unfortunately the included studies did not contain significant information about disease stages or ethnic variations that could explain this. We did discuss some interesting findings in patients with PCV for example in this section.
- The discussion effectively synthesizes findings but could benefit from more critical analysis. For example, why do some studies show significant associations between NLR and disease severity while others do not? Are there thresholds for clinical utility?
This is a very interesting point and we have added this in our conclusions as well, but as the results in most studies were driven by ROC curves the studies did not use a uniform cutoff and therefore it is hard to extrapolate the findings and suggest a cutoff for use in a clinical setting.
- The authors suggest using NLR as a prognostic marker but do not address practical challenges, such as variability in laboratory standards or the influence of comorbidities. Adding this context would improve applicability.
Response: We have added this, thank you very much for the advice.
- Future research directions are mentioned briefly. Expanding on specific gaps, such as the need for longitudinal studies or the integration of NLR with imaging biomarkers, would be helpful.
Response: Thank you very much, we have added this now.
Minor opinions:
- Page 1, Abstract: “Retina” should be “retinal” in the keywords section for consistency.
- Page 4: “Drusogenesis” should be corrected to “drusenogenesis”.
Response: This has been changed.
- Page 9: Sentence starting with "Interestingly, some studies tried to evaluate..." has redundant phrasing and could be restructured for clarity.
- Tables: Ensure uniform formatting of table headings and consistent font size. Table 1 and Table 2 headings differ in style.
- References: Ensure consistent formatting of journal titles, with some entries missing capitalization (e.g., "Ther Clin Risk Manag" vs. "Int Ophthalmol").
- Figures and Tables: Include legends that are more detailed to facilitate standalone interpretation.
- Page 3: Missing commas in multi-clause sentences, such as “Further studies are needed to elucidate the exact role of NLR in retinal diseases and its potential integration into clinical practice.”
Round 2
Reviewer 1 Report
Comments and Suggestions for Authors
Dear authors,
there has been a minor improvement, however I can still find several flaws. The abstract still does not contain relevant information regarding the search.
The material&method part is still not enough. Please search relevant articles that have been published to see how this part should be written. Inclusion and exclusion criteria could be better presented also in a table ( there are also only the exclusion criteria mentioned).
Author Response
here has been a minor improvement, however I can still find several flaws. The abstract still does not contain relevant information regarding the search.
The material&method part is still not enough. Please search relevant articles that have been published to see how this part should be written. Inclusion and exclusion criteria could be better presented also in a table ( there are also only the exclusion criteria mentioned).
Unfortunately, we cannot expand the methods section of our manuscript. We have read and published many review papers in high esteemed journals to confirm our decision.
Reviewer 2 Report
Comments and Suggestions for Authors
Thank you for addressing the previous comments and revising the manuscript. The updated version has improved clarity, organization, and discussion. The manuscript is now suitable for publication.
Comments on the Quality of English LanguageThe revised version demonstrates clear improvements in language and readability. While the English is generally correct, minor refinements in sentence structure and phrasing could further enhance clarity and conciseness. Overall, the language is sufficient to convey the research effectively.
Author Response
Thank you for addressing the previous comments and revising the manuscript. The updated version has improved clarity, organization, and discussion. The manuscript is now suitable for publication.
Thank you very much for your comments.